# Ex Vivo Treatment with Allogenic Mesenchymal Stem Cells of a Healthy Donor on Peripheral Blood Mononuclear Cells of Patients with Severe Alopecia Areata: Targeting Dysregulated T Cells and the Acquisition of Immunotolerance

**DOI:** 10.3390/ijms232113228

**Published:** 2022-10-30

**Authors:** Jung-Eun Kim, Yu-Jin Lee, Kyung-Jae Lee, Song-Hee Park, Hoon Kang

**Affiliations:** Department of Dermatology, Eunpyeong St. Mary’s Hospital, College of Medicine, The Catholic University of Korea, Seoul 03312, Korea

**Keywords:** mesenchymal stem cells, alopecia areata, peripheral mononuclear cells, T cells, allogenic

## Abstract

Alopecia areata (AA) is an autoimmune condition related to the collapse of the immune privilege of hair follicles. Certain AA populations present severe clinical manifestations, such as total scalp hair or body hair loss and a treatment refractory property. The aim of this study was to assess the effects of allogenic human mesenchymal stem cells (hMSCs) from healthy donors on the peripheral blood mononuclear cells (PBMCs) of severe AA patients, with a focus on the change in the cell fraction of Th1, Th17, and Treg cells and immunomodulatory functions. PBMCs of 10 AA patients and eight healthy controls were collected. Levels of Th17, Th1, and Treg subsets were determined via flow cytometry at baseline, activation status, and after co-culturing with hMSCs. All participants were severe AA patients with SALT > 50 and with a long disease duration. While the baseline Th1 and Treg levels of AA patients were comparable to those of healthy controls, their Th17 levels were significantly lower than those of the controls. When stimulated, the levels of CD4+IFN-γ+ T cells of the AA patients rose sharply compared to the baseline, which was not the case in those of healthy controls. The cell fraction of CD4+Foxp3+ regulatory T cells also abruptly increased in AA patients only. Co-culturing with allogenic hMSCs in activated AA PBMCs slightly suppressed the activation levels of CD4+INF-γ+ T cells, whereas it significantly induced the differentiation of CD4+Foxp3+ regulatory T cells. However, these changes were not prominent in the PBMCs of health controls. To examine the pathomechanisms, PBMCs of healthy donors were treated with IFN-γ to induce AA-like environment and then treated with allogenic grants and compared with ruxolitinib as a positive treatment control. hMSC treatment was shown to significantly inhibit the mRNA levels of proinflammatory cytokines, such as IFN-γ, TNF-α, IL-1α, IL-2R, IL-15, and IL-18, and chemokines, such as CCR7 and CCR10, in IFN-treated PBMCs. Interestingly, hMSCs suppressed the activation of JAK/STAT signaling by IFN in PBMCs with an effect that was comparable to that of ruxolitinib. Furthermore, the hMSC treatment showed stronger efficacy in inducing Foxp3, IL-10, and TGF-β mRNA transcription than ruxolitinib in IFN-treated PBMCs. This study suggests that allogenic hMSC treatments have therapeutic potential to induce immune tolerance and anti-inflammatory effects in severe AA patients.

## 1. Introduction

Alopecia areata (AA) is an autoimmune hair-loss disorder with the defect of hair cycling resulting from the collapse of hair follicle (HF)-immune privilege (IP) [1,2]. In the acute stage, characteristic peribulbar inflammatory infiltrates can be seen around HF, which is the so-called ‘swarm of bees’ sign. In this stage, interleukin (IL)-15 and interferon (IFN)-γ produced from peribulbar pathogenic CD8+ cytotoxic T cells and natural killer cells are considered to be the major pathogenesis of acute AA [3]. Meanwhile, the chronic stage of severe AA often lacks peribulbar inflammation, and little is known about its pathomechanism. Most chronic and severe AA patients do not respond to conventional immunosuppressive treatments, such as systemic corticosteroid and cyclosporin or immunotherapy with Diphenylcyclopropenone (DPCP) [4]. Recent developments and applications of Janus kinase (JAK) inhibitors for the treatment of AA achieved remarkable results [5]. Blocking the JAK signaling pathway in pathogenic T cells and natural killer T cells effectively inhibits downstream molecule IL-15 and IFN-γ and subsequentially suppresses positive feedback for expansion of pathogenic T cells. Although most AA patients in acute inflammatory stage shows dramatic responses to JAK inhibitors, certain AA patients in the chronic non-inflammatory phase do not respond to JAK inhibitors at all [6]. Moreover, even patients with good responses to JAK inhibitors require long-term administrations of the drug because AA easily relapses 2.7 months after the cessation of the drug [7]. As there are no established biomarkers of AA, which would reflect disease activities and therapeutic responses, many researchers have made substantial efforts to reveal biomarkers of AA, which patients would be indicated for JAK inhibitors, or other treatments such as DPCP [8,9]. There is some evidence that elevated serum levels of IL-4 and IL-12 before treatment suggest unfavorable response to DPCP in AA patients [8]. However, DPCP treatments often take a long period of time, ranging from several months to years until clinical improvement appears, and it may lead to unwanted side effects such as systemic contact dermatitis [10].

Unmet needs such as safety issues for long-term use of conventional treatment, lack of efficacy, long duration to remission, and unwanted side effects has led to a new paradigm in AA treatment. Stem cell therapy can be one treatment option because of its immunomodulatory effects and lack of toxicity. Mesenchymal stem cells (MSCs) are considered to have immunotolerance potential by secreting potent immunosuppressive cytokines such as IL-10 and TGF-β, and they are considered not to induce immune rejection despite the fact that they originate from other individuals. In this background, few clinical trials using allogenic stem cell therapy have been studied in several refractory diseases, including graft-versus-host diseases and rheumatoid arthritis [11,12]. Attempts to treat AA with MSCs or their conditioned medium are currently under active investigation [13,14,15,16,17]. Various origins of MSCs have been studied for hair loss treatments, including hematopoietic stem cells, hair follicle stem cells, and adipose-tissue-derived MSCs. After one-time exposure to allogenic MSCs from healthy donors by extracorporeal circulation, certain chronic AA patients showed dramatic hair regrowth, and the effects were sustained up to 1 year [14]. A single intradermal injection with allogenic Wharton-jelly MSCs showed an average hair regrowth of 67% in four AA patients during a six-month follow-up period [15]. Adipocyte-derived stem-cell-conditioned media combined with a 10,600 nm carbon dioxide fractional laser or micro-needling induced more than 50% of hair regrowth in 9 out of 14 refractory AA patients. Six of them achieved complete hair regrowth in 8–16 weeks [17]. However, the therapeutic mechanism of MSC treatment (MSCT) is not yet fully understood.

We previously studied the effects of allogenic hMSCs on various hair-comprising cells with a focus on cellular growth, anagen-inducing properties, the restoration of hair follicle immune privilege (HF-IP), and anti-inflammatory effects [18,19]. We revealed that allogenic hMSCs reverted proinflammatory responses and stimulated anagen-inducing Wnt/b-catenin-signaling molecules in both dermal papilla cells and outer root sheath keratinocytes. MSCT also significantly improved HF-IP and enhanced both the survival and growth in the length of murine vibrissa HFs ex vivo organ culture compared to those of IFN-treated HFs [18,19].

In this study, we investigated the possible therapeutic effect of allogenic hMSCs—donated from healthy young individuals—on peripheral mononuclear cells (PBMCs) of severe AA patients. To further examine the therapeutic mechanisms of MSCT, we induced an AA-like microenvironment via pretreatments with INF on PBMCs from healthy donors and treatment with allogenic hMSCs. We focused on the effects of allogenic hMSCs on the differentiation of T cell fraction and the activation of the JAK/STAT signaling pathway and several pro-inflammatory and immunosuppressive effects.

## 2. Results

### 2.1. T Cell Population Characteristics in Severe AA Patients

In total, 10 severe AA patients (6 females and 4 males) were enrolled in this study. Their average age was 43.8 ± 12.5 years and the average age of onset of AA was 33.4 ± 15.9 years old. The mean SALT score was 81.3 ± 21.7%, and the mean disease duration was 10.3 ± 7.4 years. Their demographic characteristics are listed in Table 1 and Table 2. CD4+T cells were separated from PBMCs from the patients and healthy controls, and flow cytometry was performed to examine Th1, Th17, and Treg cell subsets. Relative cell counts (%) of the total CD4+T cells relative to total lymphocytes were significantly higher in AA patients than in healthy controls. The baseline levels of IL17−IFN-γ+Th1 and IL17−Foxp3+Treg cell/total CD4+T cell ratio were not different between AA patients and healthy controls. Interestingly, IL17+IFN-γ−Th17’s cell count significantly decreased in AA patients compared to the control (Figure 1).

### 2.2. Effects of Allogenic hMSCs on T Cell Differentiation in AA Patients

hMSCs alone slightly decreased the cell viability of PBMCs of AA patients, whereas hMMSCs with IFN or IFN alone did not affect their viability, and ruxolitinib, a JAK1/2 inhibitor, significantly suppressed it at higher concentrations (Figure 2).

Next, we investigated the effect of allogenic hMSCs on T cell differentiation in AA patients and healthy controls. CD4+T helper cells in both AA and healthy control were stimulated by CD3 and CD28, which are further enhanced by co-culturing with hMSCs, but the difference between baseline and activation plus exposure to hMSCs was only statistically significant in healthy controls. Interestingly, lymphocytes from AA patients were significantly differentiated into IL-17A−IFNr+Th1 cell subsets when stimulated compared to controls (*p* < 0.05), and these changes were suppressed by hMSCs, although they were not significant. Baseline IL-17A+IFN−Th17 cell subsets were lower in AA PBMCs compared to those of healthy controls, and they were barely activated by stimulation. Meanwhile, the relative ratio of IL-17A+IFN−Th17 cells of healthy controls was variably seen at baseline and in stimulated groups, and it was less affected by hMSC treatments. The percentage of the IL-17A−Foxp3+ Treg cells of AA patients were similar to those of healthy controls at baseline levels; however, they were significantly activated by CD3 stimulation, unlike healthy controls. These activations were synergistically enhanced by allogenic hMSC exposure (Figure 3).

### 2.3. Effects of Allogenic hMSCs on the Protein Expression of JAK/STAT Pathway-Related Molecules in IFN-γ-Treated PBMCs

We examined the inhibitory effect of allogenic hMSCs on the JAK/STAT signaling pathway in IFN-treated PBMCs from healthy donor to mimic an AA-like microenvironment, and ruxolitinib was used as positive treatment control. As ruxolitinib is known to be a JAK1/2 inhibitor, it completely blocked the phosphorylation of JAK1 and 2 and partially inhibited JAK3 phosphorylation in PBMCs. The phosphorylation levels of JAK1, 2, and 3 and STAT1, 2, and 3 were effectively suppressed by ruxolitinib in IFN-pretreated PBMCs. Interestingly, the hMSC-induced suppression in the levels of phosphorylation of JAK1, 2, and 3 and STAT1, 2, and 3 in PBMCs were all comparable to those by ruxolitinib (Figure 4).

### 2.4. Effects of Allogenic hMSCs on the mRNA Expression of T Cell Differentiation and Inflammatory Markers in IFN-γ-Treated PBMCs

Genes responsible for T cell differentiation and inflammatory responses were examined at mRNA levels. IFN treatment shows a significant surge in inflammatory cytokines and chemokines, including IL-17, IFN-γ, IL-1α, IL-2 receptor γ, IL-15, IL-18, CCR7, CCR10, and TNF-α mRNA in PBMCs. Ruxolitinib significantly reverted the expression of these proinflammatory cytokines and chemokines. Similarly, hMSCs alone or with IFN significantly inhibited the IFN-induced expressional levels of IFN-γ, IL-1α, IL-2 receptor γ, IL-15, IL-18, and TNF-α in a manner comparable to ruxolitinib. However, it must be noted that IL-17 was not effectively suppressed by hMSCs when it was administered with IFN. This was seen in the same way with the ruxolitinib treatment. Interestingly, the transcription of IL-18 was significantly downregulated by hMSCs, whereas that by ruxolitinib was not.

Interestingly, the mRNA of anti-inflammatory cytokine IL-10 and TGF-β and the potent Treg cell inducer dramatically increased in PBMCs in response to allogenic MSCT, whereas they were not significantly induced by ruxolitinib. IL-10 mRNA was prominently enhanced by hMSC treatments even in the IFN-treated group. While MSCT dramatically stimulated Foxp3 and TGF-β mRNA when treated alone, it was not increased enough under IFN co-treated microenvironment. Although ruxolitinib significantly increased the expression of Foxp3, hMSCs induced Treg differentiation markers that were greater than ruxolitinib (Figure 5).

## 3. Discussion

We investigated how allogenic MSCTs could affect the differentiation and immune response of CD4+T cell subset of severe AA patients. Severe AA patients showed different baseline CD4+T cell compositions and characteristics compared to health controls (Table 3). While there were no significant differences in Th1 levels at the baseline, the Th1 cells of AA patients showed significantly higher responsiveness than those of the controls when they were stimulated. Interestingly, differentiation into IFN+ Th1 cell counts in AA patients were inhibited by co-culturing with allogenic hMSCs, although not in a significant manner. Differentiation into Foxp3+ Treg cell subsets increased by the activation stimuli and sharply increased more by exposure to allogenic MSCs in the AA group, which was not shown in the healthy control group. This means that allogenic MSCT has a therapeutic potential to suppress pathogenic Th1 cell expansion as well as strongly enhance Treg cell induction in AA when administrated systemically. Meanwhile, baseline IL17A+ Th17 cell fractions in severe AA were not only significantly lower than the control group, but the responsiveness was blunted when stimulated.

We further evaluated the effects of allogenic MSCT on the IFN-treated PBMCs of a healthy donor to simulate the PBMCs of AA patients. Most proinflammatory cytokines, including IL-17, IFN-γ, IL-1α, IL-2Rγ, IL-15, IL-18, and TNF-α, and chemokines CCR7 and CCR10 were highly enhanced by IFN treatment and reverted by allogenic hMSCs, which was similar to the results obtained with ruxolitinib at the mRNA level. At protein levels, we found that hMSCs effectively suppressed the IFN-induced expression of JAK1, 2, and 3 and STAT1, 2, and 3 in PBMCs.

The composition of cell subsets can vary depending on the disease courses of AA. Peripheral blood CD4+CD8+T cells are reported to be reduced, and the CD4/CD8 ratio is high in acute AA patients compared to healthy controls. The ratio of CD4/CD8 is proportionate to the disease duration and the SALT score [20]. Our data consistently show that CD4+T cell counts are significantly higher in chronic and severe AA patients compared to controls. Meanwhile, higher proportions of CD4+CD8+T cells are infiltrated and seen around the affected HFs in AA [20]. The expressions of cytokines and chemokines and CD8+ and CD4+T cell subpopulations are reported as being different during the disease phases in the C3H/HeJ AA mice model [21].

Because AA is considered to be an autoimmune disease, it has been well studied that Th1 cytokines increased in the serum of AA patients with their association with disease activity. However, the Th1 cell fraction in PBMCs in AA has not been studied enough. Interestingly, the numbers of CD4+IFN+Th1 cells in AA were not different compared to control, and they were found to be highly sensitive to activation stimuli compared to control in our study. There is evidence that central CD4+ memory T cells increased in a chronic AA mice model [21]. Increased serum levels of IFN-γ, IL-1β, and IL-6 have been reported in 33 acute AA patients compared to healthy controls [22].

Serum IFN-γ expression is known to be proportional to the extent of AA involvement, as alopecia totalis (AT) or alopecia universalis (AU) patients have shown significantly higher levels of IFN-γ in the serum. Serum IFN-γ has been suggested to be a biomarker reflecting the disease activity [23,24]. Indeed, large amounts of IFN-γ are produced by perifollicular T cells, and Natural Killer (NK) cells are central players of AA pathobiology [25]. The systemic or local administration of IFN-γ has also been used to induce AA in mice models [26]. We found that allogenic MSCs decreased further differentiation of naïve T cells into CD4+IFN+Th1 cells by suppressing the JAK/STAT signaling pathway.

AA patients are reported to show relative deficiency of CD4+CD25+Foxp3+Treg cell population in the serum [27]. By contrast, another study reported that serum Treg cells in severe AA is higher than in mild AA [27]. In our study, serum Treg cell population levels were not decreased at baseline levels compared to controls. This may be attributable to the difference in patients’ chronicity and the stage (active or stable) they are in. Treg cells are known to be increased in acute stage in PBMCs of AA patients and decline as it goes to the chronic stage. A recent study by Hamed, F. N. et al. revealed that CD39-expressing suppressive Treg subpopulation decreased in AA patients compared to healthy controls both in the serum and around the HFs. They investigated PBMCs in nine patch AA, five AT and six AU patients, which is similar to our study’s participants [28]. Although we did not further evaluate the subpopulation of Treg cells, we infer that allogenic MSCT would strongly induce suppressive Treg subpopulations since MSCs significantly downregulated JAK/STAT signaling and inhibited the expression of proinflammatory cytokines and chemokines in our study. We believe that acquisition of both immunotolerance and immunosuppression of pathogenic T cells by allogenic MSCT may increase the chance of achievement of complete remission and long-term maintenance.

We found that the baseline Th17 cellular fraction of severe AA patients decreased compared to healthy controls. Although detailed roles of Th17 cell in the pathogenesis of AA are not known, according to a study by Han YM et al., levels of circulating Th17 cells increased primarily in the early stage or active phase AA and negatively correlated with the duration of the disease [27]. Our low Th17 cell fractions are consistent with those reported by Han YM et al. because our participants had a long disease duration, mostly of more than 3 years. Interestingly, in this study, MSCT affected cellular fraction of Th17 cells less in both AA patients as well as in controls.

MSCs are known to induce Treg cell differentiation by secreting large amounts of TGF-β and IL-10 [29]. We found that allogenic MSCs induced naïve T cells to differentiate into T reg cells in AA patients. Consistent with that, the transcription to Foxp3, TGF-β2, and IL-10 in PBMCs significantly increased in co-culturing with MSCs. However, only Treg cellular subsets from severe AA patients dramatically increased after 3 days of exposure to allogenic MSCs, whereas those from healthy controls did not. This suggests that interactions between allogenic MSCs and recipient central immune cells only occurs under immunologically abnormal statuses and that they do not induce specific immune responses under normal healthy conditions.

Because the JAK/STAT signaling pathway in pathogenic T cells is understood to be the major pathogenesis of AA [5,30], we investigated whether MSCT could inhibit JAK/STAT signaling and inflammatory cascades in the peripheral blood T cells using ruxolitinib as a positive treatment control. MSCT significantly blocked the phosphorylation of JAK1,2, and 3 plus STAT 1,2, and 3, which is comparable to the effects of ruxolitinib. Although MSCT affects multiple targets and does not selectively target JAK/STAT signaling pathway, their inhibitory effects on JAK/STAT signaling were not inferior to those of ruxolitinib. A recent study reported that hematopoietic cells more selectively express JAK3 signaling, whereas JAK1/2 signaling was more broadly expressed in other cell types. Blocking the JAK3 pathway is necessary and sufficient to prevent and reverse AA in the C3H/HeJ mice model [30]. In a promising finding, the capacity of MSCs for inhibiting JAK3 signaling is remarkable in our study. In addition to blocking JAK/STAT signaling molecules, we found that the transcription of IFN signature genes and other proinflammatory cytokines and chemokines were significantly suppressed by MSCT in IFN-treated PBMCs.

Therapeutic potentials of MSCT in AA have been consistently suggested, but the lack of clinical data is an obstacle to its progress with respect to clinical applications in the real world. The assessment of the safety and effectiveness of allogenic MSCT should be based on personalized medicine, as therapeutic outcomes are determined by donor and host factors. These ex vivo co-culture models can be utilized to predict interactions between allogenic MSCs and PBMCs of AA recipients by considering various immunologic mechanisms between individuals. It should be studied to identify and administer an effective substance to induce immunotolerance from AA-specific secretome for the future. This would allow allogenic MSCT to become a standardized therapeutic option.

The limitation of our study is that blood sampling and analysis was performed only at one time point of the disease course in a small number of patients, and the hMSCs used in this study were obtained from one healthy donor. Comparison analyses with serial blood sampling would help prove the therapeutic effects of MSCT in the future. Although serum biomarkers have been suggested to reflect disease activities, PBMCs may not completely represent immune processes of Th1, Th17, and Treg cells in AA-affected scalp tissue. Interactions between hMSCs and T cells in the AA scalp tissue should be studied to confirm this.

## 4. Materials and Methods

### 4.1. Patient Selection and Blood Samples

All patients were diagnosed with AA at the Catholic University of Korea, Eunpyeong St. Mary’s Hospital. The diagnosis of AA was made based on clinical features, and other possible differential diagnoses—such as androgenic alopecia or other type of hair loss—were carefully excluded. Consenting AA patients who fulfilled the following three criteria were included in this study: (i) patients with severe AA with SALT score ≥ 50%, (ii) patients in treatment-refractory chronic stage with long disease duration, and (iii) patients free of any other comorbid autoimmune or atopy diseases, such as psoriasis, vitiligo, autoimmune thyroiditis, rheumatoid arthritis, asthma, atopic dermatitis, allergic rhinitis, or urticaria. Patients who received any systemic immunosuppressants such as corticosteroid or cyclosporine for at least 1 month were excluded.

Healthy volunteers free of any systemic or chronic inflammatory diseases participated as controls. Heparinized peripheral blood (50 mL) was collected from 8 healthy volunteers as well as from 10 patients with AA. All procedures were carried out in accordance with the Institutional Review Board and Use Committee’s approved protocols (IRB no. PC17TESI0034 and PC22TISI0135).

### 4.2. PBMC Isolation

PBMCs were isolated from heparinized peripheral blood samples using Ficoll-Histopaque 1077 (Sigma-Aldrich, St. Louis, MO, USA) density gradient solution in a previously described method [31]. The cells were cultured in RPMI 1640 (Gibco, Grand Island, NY, USA) supplemented with 10% FBS (FBS; Gibco BRL, Life Technology, Karlsruhe, Germany) and 1% penicillin/streptomycin (Gibco, BRL, Life Technology, Karlsruhe, Germany) with the addition of stimulatory agents.

### 4.3. hMSC Culture

We obtained human bone marrow-derived MSCs (Catholic MASTER Cells from the Catholic Institute of Cell Therapy (CIC, Seoul, Korea). Allogenic MSCs that had been donated from a healthy, young donor in the GMP facility were produced to prevent possible immune reactions and transmissible infectious diseases. The hMSCs in passages 3 to 5 were used for the experiments. hMSCs were seeded at a density of 5 × 10^4^ cells per well for 24 h in the polyethylene terephthalate-coated upper chamber (pore size: 0.8 mm, 24-well format, Cell Culture Inserts; Corning, Falcon, Franklin Lakes, NJ, USA) of the Transwell culture plates (Cell Culture Insert Companion Plate; Corning, Falcon, Franklin Lakes, NJ, USA).

The culture medium was Dulbecco’s Modified Eagle Medium (DMEM high glucose, Gibco BRL, Life Technology, Karlsruhe, Germany) containing 10% fetal bovine serum (FBS; Gibco BRL, Life Technology, Karlsruhe, Germany) and 1% penicillin/streptomycin (Gibco, BRL, Life Technology, Karlsruhe, Germany). After 24 h, the upper chamber containing hMSCs was transferred to the wells for co-culture with PBMCs.

### 4.4. T Cell Preparation and Co-Culture with hMSCs

Isolated human PBMCs were plated into 24-well culture plates at a density of 5 × 10^5^ cell/mL per well and cultured in RPMI 1640 (Gibco, Grand Island, NY, USA) containing 10% FBS, IL-2 (1 μg/mL, R&D systems, Minneapolis, MN, USA), and anti-CD28 (1 μg/mL, eBioscience, San Diego, CA, USA) monoclonal antibody. On the day prior to plating, 24-well culture plates were coated with anti-CD3 (1 μg/mL, eBioscience, San Diego, CA, USA) monoclonal antibody. For co-culturing with hMSCs, the upper chambers containing hMSCs were transferred to the lower chambers, where PBMCs were then cultured and incubated for 3 days (72 h) at 37 °C with 5% CO_2_ (Figure 6).

### 4.5. Flow Cytometry

For the flow cytometry analysis, Cell Stimulation Cocktail (2 µL/mL, eBioscience, San Diego, CA, USA) and Protein Transport Inhibitor (2 µL/mL, eBioscience, San Diego, CA, USA) were both added to the ‘unstimulated’ group. After 6 h of cell stimulation, PBMCs were collected and resuspended in Flow Cytometry Staining Buffer at the concentration of 5 × 10^5^ cells per 100 µL. Cells were stained with CD4-FITC (1 µL/100 µL, eBioscience, San Diego, CA, USA) and Viability Dye eFluor™ 780 (1 µL/100 µL, eBioscience, San Diego, CA, USA) and incubated at 4 °C for 30 min in the dark. After being incubated, cells were washed in a staining buffer and fixed with fixation buffer at room temperature for 30 min in the dark. The cells were washed in a Permeabilization Buffer and stained with IFN-γ-PE, IL-17A-APC, Foxp3-PerCP-Cyanine5.5 (1 µL/100 µL each, eBioscience, San Diego, CA, USA) at 4 °C for 1 h in the dark. Flow cytometry analyses were conducted using FACS Canto (BD, San Diego, CA, USA). Data were analyzed using FlowJo software (Treestar, Inc., San Carlos, CA, USA).

### 4.6. In Vitro AA Modeling with PBMCs from a Healthy Donor

We obtained PBMCs from a healthy donor with no history of systemic or chronic inflammatory diseases. Human PBMCs were plated into 24-well culture plates at a density of 5 × 10^5^ cell/mL per well and cultured in RPMI 1640 (Gibco, Grand Island, NY, USA) containing 10% FBS and IL-2 (1 μg/mL, R&D systems, Minneapolis, USA).

An AA-like microenvironment was induced by treatments with recombinant human IFN-gamma—which was obtained from Peprotech (315-05, Rocky Hill, NJ, USA)—on PBMCs at 100 ng/mL. Ruxolitinib measuring 10 nM/mL was used as a positive treatment control. For the MSC co-cultured group, the upper chamber containing hMSCs (5 × 10^4^ cell per well) was transferred to the lower chamber, where PBMCs were cultured to produce the co-culture for 3 days. After 3 days of incubation at 37 °C with 5% CO_2_, the upper chamber was removed so that PBMCs could be harvested for analysis.

### 4.7. Cell Viability Assay

We performed a WST-8 [2-(2-methoxy-4-nitrophenyl)-3-(4-nitrophenyl)-5-(2,4-disulfophenyl)-2H-tetrazolium, monosodium salt] assay to assess cell viability. We used CCK-8 KIT (Dojindo Molecular Technologies, Rockville, MD, USA) to determine the viability of PBMCs at various concentrations of IFN-γ and ruxolitinib.

Briefly, PBMCs (5 × 10^3^ cells/well) were seeded into 96-well plates. After 3 days of incubation with IFN-γ (50 ng/mL, 100 ng/mL, 500 ng/mL) or ruxolitinib (1 nM, 10 nM, 100 nM), the CCK-8 solution was added and reacted for 4 h. Then, samples were taken to measure absorbance at 450 nm using an enzyme-linked immunosorbent assay plate reader.

### 4.8. Real-Time PCR

Following the manufacturer’s recommendations, cDNA was synthesized using the MG cDNA Synthesis Kit (CancerROP, Seoul, Korea) and total RNA was extracted from PBMCs using TRIzol reagent (Invitrogen, Carlsbad, CA, USA). Total RNA (1 μg) was reverse transcribed with primers and Moloney-murine leukemia virus (M-MLV) reverse transcriptase (RTase) (CancerROP, Seoul, Korea). SYBR Green Master Mix (CancerROP, Seoul, Korea) was used for real-time PCR. The levels of gene expression were measured using analysis software (Quantity One 1-D analysis, Bio-Rad, Hercules, CA, USA). Appendix A lists the primer sequences. The primers were designed and obtained from Bioneer (Bioneer, Daejeon, Korea).

### 4.9. Western Blot Analysis

RIPA lysis buffer along with Protease and Phosphatase Inhibitor Cocktail (ThermoFisher, Rockford, IL, USA) was used to collect PBMCs. The amount of protein was determined using the BCA protein assay kit (ThermoFisher, Rockford, IL, USA) and compared to standards of BSA (ThermoFisher, Rockford, IL, USA). Electrophoresis was used on sodium dodecyl sulfate (SDS)-polyacrylamide gels to separate cell lysates with the same amount of total protein, and iBlot (ThermoFisher, Rockford, IL, USA) was used to transfer the separated samples to polyvinylidene fluoride (PVDF) membranes.

The membranes were incubated with primary antibodies in 5% BSA at 4 °C overnight after being blocked with 5% BSA in TBST for an hour at room temperature. B-actin was obtained from Santa Cruz Biotechnology (Santa Cruz, CA, USA), and the primary antibodies against phospho-JAK1-3, total-JAK1-3, and phospho-STAT1-3 total-STAT1-3 were obtained from Cell Signaling Technology, Inc. (Cell Signal, Beverly, MA, USA). The membranes were incubated with peroxidase-conjugated secondary antibody (Cell Signaling Technology, Beverly, MA, USA) in 5% BSA for 1 h at room temperature after being repeatedly washed with TBST the following day. The immunoreactive bands were detected using an enhanced chemiluminescence (ECL) detection system (ThermoFisher, Rockford, IL, USA) and obtained using a chemiluminescence imaging system (Chemi-Doc Imaging System; Biorad, Hercules, CA, USA) after being incubated with horseradish peroxidase.

### 4.10. Statistical Analysis

All data are expressed as mean ± SEM. Student’s t-test was used for comparisons between two groups and one-way ANOVA was used to compare among three groups. All tests were one-sided, and a *p*-value of less than 0.05 was considered to be statistically significant. The data were statistically analyzed using GraphPad Prism software (San Diego, CA, USA).

## 5. Conclusions

AA is an autoimmune hair loss disorder characterized by the loss of the immune privilege of HFs and peribulbar inflammation in the affected scalp area. Although recent advances in the understanding of JAK/STAT signaling in local infiltrates of pathogenic T cells have led to the development and introduction of JAK inhibitors in the treatment of AA, the pathogenesis of chronic and severe forms of AA is still enigmatic, resulting in the absence of effective treatments. We presumed that AT/AU patients would have systemic immune abnormalities and found that they had low baseline levels and blunted responsiveness in Th17 cells, but high responsiveness in Th1 and Treg cells, unlike healthy controls. Allogenic MSCs on PBMCs of severe AA patients were able to strongly induce Treg cell differentiation without significantly impairing Th1 or Th17 cellular fractions. From these findings, we suggest that MSCT would immunomodulate pathogenic T cells of AA without any suppression of general immunity in an ex vivo culture condition. Our data revealed that allogenic MSCs reverted the INF-induced expression of proinflammatory cytokines and chemokines and enhanced immunotolerance-related cytokines in PBMCs. Taken together, our data to understand immunomodulatory effects and the high immunotolerogenic potential of allogenic MSCT may facilitate the achievement and maintenance of complete remission in severe AA.

## Figures and Tables

**Figure 1 ijms-23-13228-f001:**
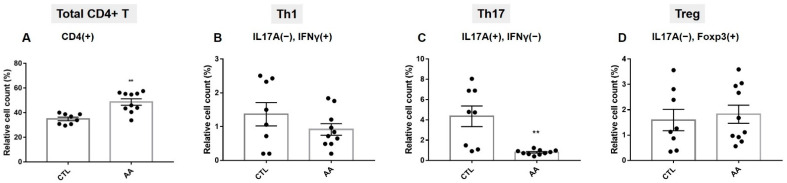
Baseline levels of total CD4+ Th cells, Th1, Th17, and Treg cell subsets in healthy controls (*n* = 8) and AA (*n* = 10). Relative cell count (%) of (**A**) total CD4+T helper cells, (**B**) IL17A−IFN-γ+ (Th1), (**C**) IL17A+IFN-γ− (Th17), and (**D**) IL17A−FoxP3+ (Treg) cells from the control group and AA group. Total CD4+ cells ratio was increased in the AA group compared to the control group. Th17 cell fractions in the AA group were all lower at baseline compared to the control group, although Th1 and Treg cell fraction showed no significant differences. The data represent the means ± SEM, statistically significant at ** *p* < 0.01 compared to the control.

**Figure 2 ijms-23-13228-f002:**
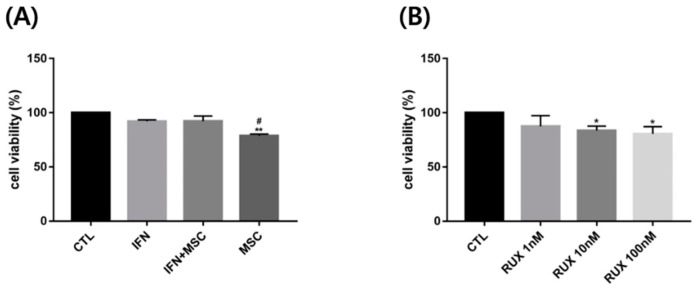
Cell viability of PBMCs from AA patients in response to treatment with hMSCs: (**A**) Co-culturing with hMSCs significantly decreased cell viability of PBMCs in AA patients compared to control and IFN-γ-treated group. (**B**) Ruxolitinib significantly suppressed the cell viability of PBMCs at concentrations above 10 nM. The data represent the means ± SEM, *n* = 3, statistically significant at * *p* < 0.05, ** *p* < 0.01 compared to control, and # *p* < 0.05 compared to the IFN-γ- treated group.

**Figure 3 ijms-23-13228-f003:**
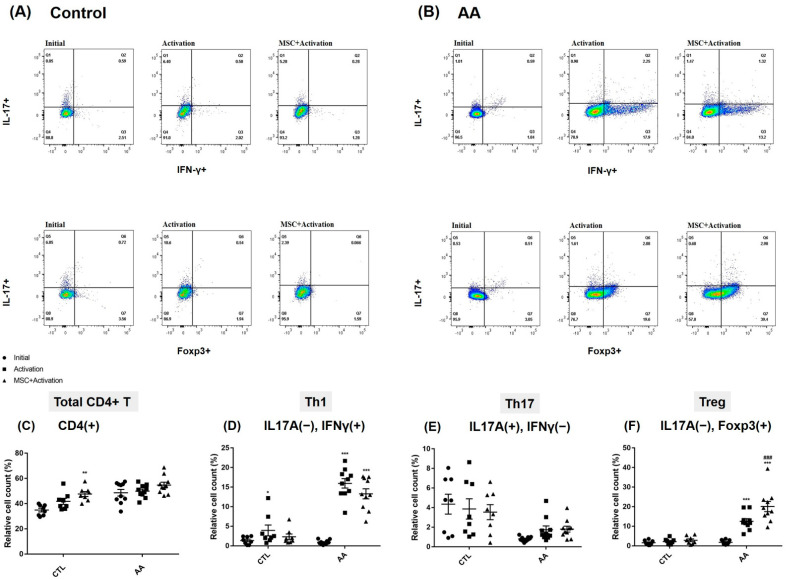
Representative FACS analysis results of IL-17+/−IFN-γ+/−cells and IL-17+/−Foxp3+/− cells at baseline, after activation, and after co-culturing with hMSCs in activation in (**A**) healthy control and (**B**) AA. The effects of hMSC treatment on (**C**) total CD4+Th, (**D**) Th1, (**E**) Th17, and (**F**) Treg cell subsets in PBMCs were examined. When stimulated, the levels of Th1 and Treg cells of AA patients rose sharply compared to baseline, whereas the cells of healthy controls rose slightly. Co-culturing with hMSCs slightly decreased the activation levels of Th1 cells compared to the activated control in AA patients. Meanwhile, Treg cells were markedly induced by allogenic hMSC treatments in AA patients compared to the activated control, although these changes were not prominent in health controls. The Th17 cell ratio was less affected by stimulation in both patients and controls. Single cell suspensions were prepared from healthy controls (CTL; *n* = 8) and alopecia areata (AA; *n* = 10) patients. The data represent the means ± SEM. * *p* < 0.05, ** *p* < 0.01, and *** *p* < 0.001 compared to initial, ### *p <* 0.001 compared to the activation group.

**Figure 4 ijms-23-13228-f004:**
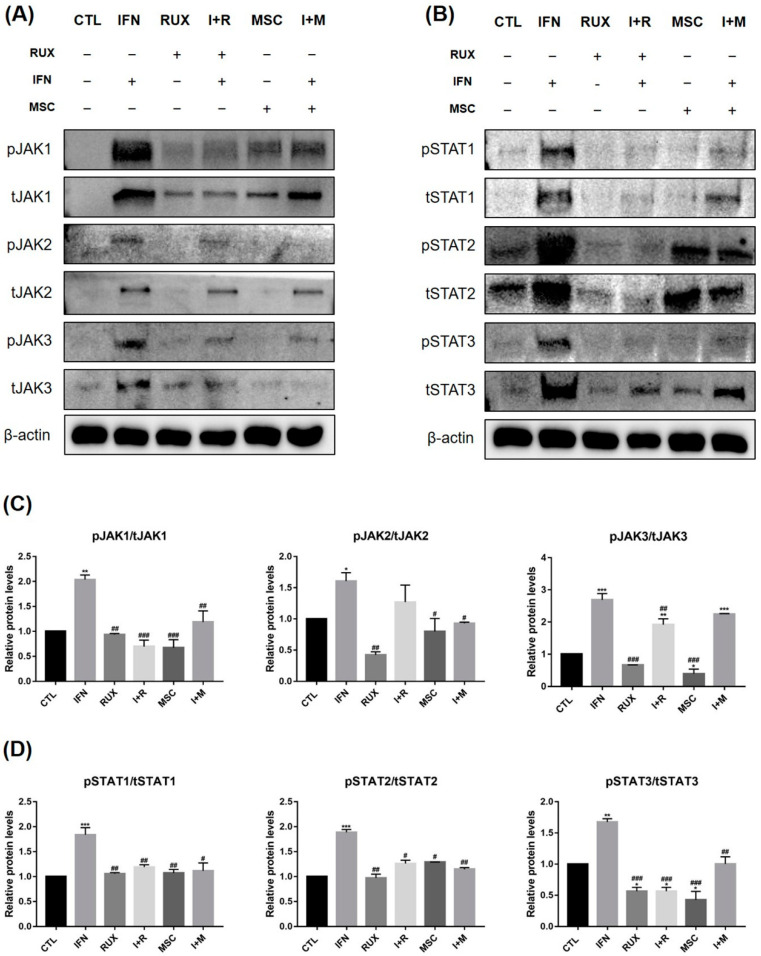
Changes in JAK/STAT pathway-related molecules in IFN-treated PBMCs after treatment with hMSCs or ruxolitinib. (**A**,**B**) Western blotting and (**C**,**D**) relative fold changes show that IFN treatment increase the levels of phosphorylated JAK1, JAK2, and JAK3 and STAT1, STAT2, and STAT3. Activated signaling of JAK/STAT pathway by IFN was remarkably suppressed by hMSC treatment at similar levels as those by ruxolitinib. The bands indicate serial protein expression levels up to 72 h after ruxolitinib treatments. The data represent the means ± SEM, *n* = 3, statistically significant at * *p* < 0.05, ** *p* < 0.01, and *** *p* < 0.001 compared to the control and # *p* < 0.05, ## *p* < 0.01, and ### *p* < 0.001 compared with the IFN-γ-treated group.

**Figure 5 ijms-23-13228-f005:**
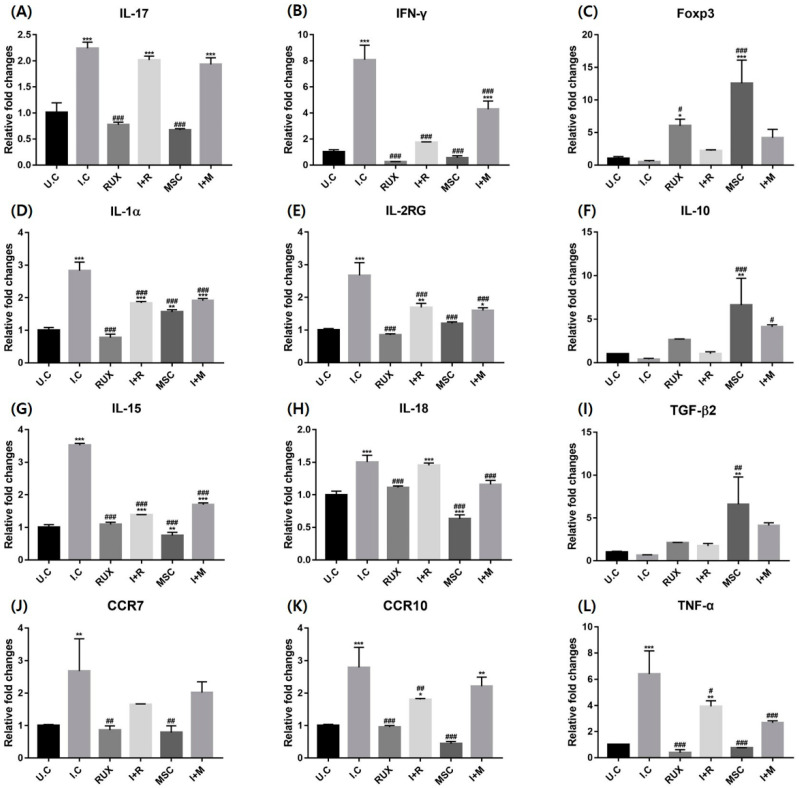
Effects of allogenic hMSC treatment on molecules related to inflammatory markers in IFN-treated PBMCs at the mRNA level. The mRNA expression of (**A**) IL-17, (**B**) IFN-γ, (**C**) Foxp3, (**D**) IL-1α, (**E**) IL-2Rγ, (**F**) IL-10, (**G**) IL-15, (**H**) IL-18, (**I**) TGF-β, (**J**) CCR7, (**K**) CCR10, and (**L**) TNF-α in PBMCs. Ruxolitinib suppressed the mRNA expression of IFN-signature molecules in PBMCs. hMSCs reverted the level of cytokines and chemokines, which was increased by IFN-γ in PBMCs. The data represent the means ± SEM, *n* = 3, statistically significant at * *p* < 0.05, ** *p* < 0.01, *** *p* < 0.001, compared to the untreated control (UC) and # *p* < 0.05, ## *p* < 0.01, ### *p* < 0.001 compared to the IFN-γ-treated control (IC). Rux: ruxolitinib; I + R: IFN-γ + Rux; MSC: hMSC-treated group; I + M: IFN-γ + hMSCs.

**Figure 6 ijms-23-13228-f006:**
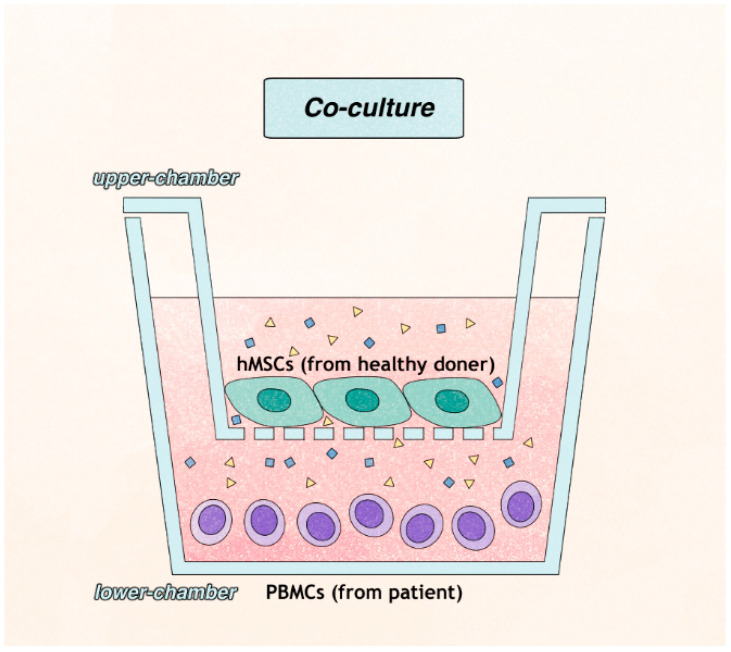
Ex vivo model for assessing the effects of hMSC treatment on hPBMC culture.

**Table 1 ijms-23-13228-t001:** Demographics and characteristics of patients included in the study.

	Refractory AA Group (*n* = 10)	Control Group (*n* = 8)	*p*-Value
Gender			0.138
Female	6 (60)	2 (25)	
Male	4 (40)	6 (75)	
Age, years	43.8 ± 12.5	32.0 ± 4.8	0.018
AA onset age, years	33.4 ± 15.9		
Disease duration, months	123.2 ± 88.4		

AA = Alopecia areata. Data are presented as mean ± standard deviation or as a number (percentage).

**Table 2 ijms-23-13228-t002:** Clinical characteristics of AA patients.

No.	Sex	Age, Years	SALT, %	AA Onset Age, Years	Disease Duration, Months	Previous Treatment for AA *						
						Systemic Steroid **	Cyclosporine **	ILI	DPCP	Oral Minoxidil	Topical Minoxidil	TCS
AA 1	F	58	53.7	52	72	-	-	-	-	-	-	-
AA 2	M	36	59	15	252	-	-	+,6 mo	-	-	+,6 mo	-
AA 3	F	25	100	18	84	+,12 mo	+,12 mo	-	+,24 mo	-	-	-
AA 4	M	37	100	35	17	-	-	+,2 mo	+,1 mo	-	-	+
AA 5	F	63	90	60	35	-	+,1 mo	-	+,2 mo	-	+	-
AA 6	F	34	100	17	204	-	-	-	+,84 mo	-	-	-
AA 7	M	37	96	33	40	-	-	+,14 mo	-	-	-	-
AA 8	M	55	50	44	132	-	-	-	-	+,24 mo	-	+
AA 9	F	53	63.8	41	144	-	+,5 mo	+,2 mo	+,5 mo	-	-	+
AA 10	F	40	100	19	252	-	-	-	+,12 mo	-	+	-

AA = Alopecia areata; ILI = triamcinolone intralesional injection; DPCP = diphenylcyclopropenone; TCS = topical corticosteroid. * “+,6 mo” means treatment in the last 6 months, “+” means treatment that is not continuous or of which the exact duration is unknown, and “-” means no treatment in the last two years. Here, Patient 1 has been without any treatment for the past 2 years. ** In the case of treatment with either systemic steroid or cyclosporine, there was no history of treatment at least 1 month prior to study participation.

**Table 3 ijms-23-13228-t003:** Baseline levels of CD4+T cellular subsets and their responsiveness and changes by co-culturing with allogenic hMSCs.

CELL SUBSET	Baseline Level	Responsiveness to Stimuli	Immunomodulation after Allogenic hMSCs
Th1	(-)	↑	↓
Th17	↓	(-)	↓
Treg	↓	↑	↑↑

hMSCs: human mesenchymal stem cells; (-): no change; ↓: decreased; ↑: increased; ↑↑: highly increased

## Data Availability

The data that support the findings of this study are available from the corresponding author upon reasonable request.

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
