# Peer review of "Ex Vivo Treatment with Allogenic Mesenchymal Stem Cells of a Healthy Donor on Peripheral Blood Mononuclear Cells of Patients with Severe Alopecia Areata: Targeting Dysregulated T Cells and the Acquisition of Immunotolerance"

_ijms, 2022, doi:10.3390/ijms232113228_

Round 1
Reviewer 1 Report
The authors presented data of significant interest to the community delineating the effects of allogenic stem cell therapy on AA particularly on PBMCs and inflammation. The authors may address the concerns mentioned below before considering for publication.
The authors failed to describe the novelty of the model as they presented the co-culture model as highlight of the work including in title & discussion. There are multiple co-culture models existing in the field and even more interesting models such as multi-layer models or organoid models which was previously developed to study AA by the same group themselves.
The authors may need to change the title and parts of the manuscript highlighting the results of the work than the model as such.
Also, as therapeutic potential of MSCs (mouse models by Weiyue Deng et al) was previously studied by other groups the authors may need to show potential implication of immune tolerance by hMSCs in pathology reduction using existing models.
Further, a larger patient sample number may help understand multiple T cell population percentage variation from onset to disease.
Author Response
We appreciate your careful review of this manuscript and have hopefully addressed the concerns to your satisfaction, as detailed below.
Point 1 : The authors failed to describe the novelty of the model as they presented the co-culture model as highlight of the work including in title & discussion. There are multiple co-culture models existing in the field and even more interesting models such as multi-layer models or organoid models which was previously developed to study AA by the same group themselves.
Response 1 : We appreciate the valuable comments. We have removed the description about establishment of co-culture models. According to the reviewer's opinion, the paper was revised to emphasize the novelty of this study.
Point 2 : The authors may need to change the title and parts of the manuscript highlighting the results of the work than the model as such. Also, as therapeutic potential of MSCs (mouse models by Weiyue Deng et al) was previously studied by other groups the authors may need to show potential implication of immune tolerance by hMSCs in pathology reduction using existing models.
Response 2 : Thank you for the comments. We changed to the tile from <Ex vivo model to assess the therapeutic effects of Allogenic Stem Cell Therapy in Alopecia areata: co-culture system of peripheral blood mononuclear cells of AA patients and human mesenchymal stem cells of healthy donors> to <Ex vivo treatment with allogenic mesenchymal stem cells of a healthy donor on peripheral blood mononuclear cells of patients with severe alopecia areata: targeting of dysregulated T cells and acquisition of immunotolerance>. To highlight our results, we revised the discussion section and added the conclusion section. In addition to therapeutic potential of MSCs, we emphasized that allogenic MSCT could raise a potential curability in chronic, and severe AA patients by simultaneously having immunosuppressive effects on pathogenic T cells and strong immunotolerance inducing effects.
Point 3 : Further, a larger patient sample number may help understand multiple T cell population percentage variation from onset to disease.
Response 3 : We totally agree with the reviewer’s opinion. The ratio of Th1, Th17, Treg cell subsets have been variably reported in previous studies [Reference 20, 27 & 28]. We think this may result from individual differences in immunological mechanism, but differences may also occur in differences in timing of blood sampling (disease courses from the onset). Next, we will study the effects of allogenic MSCT through a larger number of patients samples with different time points. We added this as our limitation in discussion section.

Reviewer 2 Report
Review of Manuscript ID: ijms-1978085, by J.E. Kim. et al., entitled “Ex vivo model to assess the therapeutic effects of Allogenic Stem Cell Therapy in Alopecia areata: co-culture system of peripheral blood mononuclear cells of AA patients and human mesenchymal stem cells of healthy donors” that is intended for publication in International Journal of Molecular Sciences
(The PDF file as Reviewer Attachment for Manuscript ID ijms-1978085 IJMS 12th October 2022 has also been added)
The subject of the manuscript is human Alopecia areata (AA), the causes of which are not fully investigated. But, it is widely believed that immune factors play a direct role in this condition. The authors proposed an interesting ex vivo model to test the regulatory effects of allogeneic hematopoietic/mesenchymal stem cells (hMSCs) derived from healthy donor on the functional properties of peripheral blood mononuclear cells (PBMCs; subpopulations of T cells such as Th1, Th17 and Treg) derived from the patients afflicted with AA. The two-chamber co-culture system proposed by the Authors for both cell types enabled to demonstrate the immunomodulatory activity of hMSCs, which is expressed, among others, by mRNA inhibition of selected pro-inflammatory cytokines in the Th1 and Th17 helper lymphocyte subpopulations. In addition, the Authors showed that in T cells experimentally stimulated with interferon-γ for upregulation of the level of cytokines and chemokines, this upward trend could be reversed with the participation of allogeneic hMSCs. Considering the applicability of hMSCs that results from their positive therapeutic effects demonstrated by the Authors, the current work is valuable, well-written in English and methodologically correct. Moreover, a wide range of the results obtained is presented in an attractive graphic form. Additionally, the Authors used adequate statistical methods to analyze the obtained results, which allowed for their thorough interpretation and critical evaluation in the context of existing knowledge.
However, in my opinion, the following points should be corrected or added before the final acceptance of the manuscript for publication, as indicated below:
1) Please clearly specify the type of stem cells used in the experiment, whether they were hematopoietic stem cells or mesenchymal stem cells, or a mix population of both cell types. The information in the Materials and Methods section indicates that they were human mesenchymal stem cells. However, in the abstract as well as in the Abbreviation section, the abbreviation hMCS was incorrectly expanded as hematopoietic mesenchymal stem cells, while the correct expansion of this abbreviation should be human mesenchymal stem cells.
2) In Table 3, in the first column (Cell subset), instead of T helper1 and T helper 17, it would suffice to describe Th1 and Th17.
3) Please add a separate Conclusions section.
4) Please adapt the format of the References section to the requirements of the International Journal of Molecular Sciences (Please use the abbreviated forms of journal titles).
In conclusion, I recommend this manuscript for publication in International Journal of Molecular Sciences, provided that the above-mentioned minor remarks and suggestions pointed out by the Reviewer will have been taken into consideration by the Authors to the re-edited and resubmitted version of current paper.

Author Response
Thank you for your thorough review.
Point 1: Please clearly specify the type of stem cells used in the experiment, whether they were hematopoietic stem cells or mesenchymal stem cells, or a mix population of both cell types. The information in the Materials and Methods section indicates that they were human mesenchymal stem cells. However, in the abstract as well as in the Abbreviation section, the abbreviation hMCS was incorrectly expanded as hematopoietic mesenchymal stem cells, while the correct expansion of this abbreviation should be human mesenchymal stem cells.
Response 1: Based on your opinion, the term and abbreviation have been corrected to the human mesenchymal stem cell used in the experiment so as not to confuse the types of stem cells.
Point 2: In Table 3, in the first column (Cell subset), instead of T helper1 and T helper 17, it would suffice to describe Th1 and Th17.
Response 2: We modified the cell subset terms in Table 3 to Th1 and Th17.
Point 3: Please add a separate Conclusions section.
Response 3: We have added Conclusions section.
Point 4: Please adapt the format of the References section to the requirements of the International Journal of Molecular Sciences (Please use the abbreviated forms of journal titles).
Response 4: We have revised the journal titles in the References section according to the format required by the International Journal of Molecular Sciences.

Reviewer 3 Report
The work “Ex vivo model to assess the therapeutic effects of Allogenic Stem Cell Therapy in Alopecia areata: co-culture system of pe-ripheral blood mononuclear cells of AA patients and human mesenchymal stem cells of healthy donors” from Kim and colleagues aims to evaluate the immunomodulatory and anti-inflammatory effects of treatment with allogenic hMSC from healthy donors on PBMCs from severe AA patients.
The work is well designed and the results convincing. I have only minor comments:
- Is it unclear the number of co-culture established.
- In figure 1 caption the wrote that the number of experiments is 2 (n=2) but this sentence does not correspond with the dots in the graphs. Is that an error? If not, the statistical analysis is not possible between two samples.
- Where are data of PBMCs viability? Lines 132-134
- In the caption of the figure 2, the n of experiments and the information about expression of the data are missing.
Author Response
Thank you for your thorough review.
Point 1: Is it unclear the number of co-culture established.
Response 1: Co-culturing with PBMCs and hMSCs for FACS analysis was done in 10 AA patients (n=10) and 8 healthy controls (n=8). For western blotting and gene expression, the number of co-culture with hMSCs and IFN-r-treated PBMCs was three (n=3). We added this content in all legends of figures.
Point 2: In figure 1 caption the wrote that the number of experiments is 2 (n=2) but this sentence does not correspond with the dots in the graphs. Is that an error? If not, the statistical analysis is not possible between two samples.
Response 2: We modified the Figure 1 description to “Figure 1. Baseline levels of Total CD4+ Th cells, Th1, Th17, and Treg cell subsets in healthy controls(n=8) and AA(n=10)” and "n=2" has been deleted.
Point 3: Where are data of PBMCs viability? Lines 132-134
Response 3: Thank you for the comment. We have added PBMC viability data to the manuscript as Figure 2.
Point 4: In the caption of the figure 2, the n of experiments and the information about expression of the data are missing.
Response 4: We appreciate the valuable comment. We added "Single cell suspensions were prepared from healthy controls (CTL; ​​n = 8) and alopecia areata (AA; n = 10) patients. The data represent the means ± SEM" to the figure.
